# Effectiveness of the Booster Dose in Protecting against COVID-19, Colombia 2022

**DOI:** 10.3390/vaccines11091461

**Published:** 2023-09-07

**Authors:** Jubby Marcela Gálvez, Ángela María Pinzón-Rondón, Henry Mauricio Chaparro-Solano, Hanna Valentina Tovar-Romero, Juliana Ramírez-Prieto, Sergio Andrés Ortigoza-Espitia, Ángela María Ruiz-Sternberg

**Affiliations:** 1Genuino Research Group, Gencell Pharma, Bogotá 111221, Colombia; 2Clinical Investigation Group, Universidad del Rosario, Carrera 24 #63C-69, Bogotá 111221, Colombia

**Keywords:** COVID-19, booster dose, effectiveness, Colombia, vaccination strategy

## Abstract

Vaccination has proven to be one of the most effective strategies against the COVID-19 pandemic. Several studies have evaluated and confirmed its effectiveness in different populations, particularly in reducing severe outcomes such as hospitalization and death. Some studies have investigated the effectiveness of vaccination against the infection, identifying the need for booster doses. This study aimed to explore the effectiveness of the vaccination schedule on the probability of infection in a sample of Colombian patients during the fourth wave of the COVID-19 pandemic, which was associated with the emergence and predominance of the Omicron variant. A cross-sectional study was conducted on individuals who underwent RT-PCR testing for COVID-19 detection in a dedicated laboratory in Bogotá, Colombia, between 30 December 2021 and 7 February 2022. A total of 1468 subjects was included in the study, of whom 36.6% (n = 538) had a positive PCR test for COVID-19. The comparison between fully vaccinated individuals with a booster dose and those without the booster dose revealed a 28% reduction in the odds of infection (OR = 0.719 CI 0.531–0.971). Age (OR = 1.009 CI 1.001–1.018) and low economic status (OR = 1.812 CI 1.416–2.319) were associated with an increased risk of infection. These findings suggest the need for a booster vaccination in the general population to improve the prevention rates of SARS-CoV-2 infection and mitigate severe outcomes.

## 1. Introduction

The COVID-19 pandemic has emerged as one of the largest and most significant public health threats that medicine has faced in the 21st century.

Efforts to contain and mitigate SARS-CoV-2 infection, a novel virus characterized by high infection and fatality rates, together with unpredictable behavior, have presented considerable challenges. A wide range of social, economic, and political factors has played a crucial role in the implementation and development of strategies against this infection [1].

The global medical and scientific community has devoted substantial efforts to identifying primary and secondary prevention measures, aimed at reducing infection and minimizing morbidity and mortality. These efforts have included the development of different treatments, such as novel antivirals, serums, and vaccines targeting COVID-19. Vaccines have been based on multiple platforms, including traditional approaches employing inactivated viruses, subunit protein-based methods, viral vector-based platforms, and innovative mRNA-based technologies. Several of these vaccines have been granted emergency use authorization and have demonstrated significant efficacy in preventing infection, severe illness, ICU admission, and death [1,2,3].

Colombia has been severely affected by the COVID-19 pandemic. As of 21 June 2023, there have been 6,371,090 confirmed cases with 142,794 deaths, reported to the WHO [4].

When we trace the epidemiological trajectory and genomic characterization of variants that emerged and became predominant in Colombia during the pandemic, four waves can be identified to date, coinciding with the identification of some WHO-designated variants of concern (VOCs). The first wave peaked in August 2020, the second wave peaked in January 2021, the third wave, the most severe and prolonged, associated with the Delta variant and the emergence of the Mu variant (first identified in Colombia by our group) [5], peaked in June 2021, and the fourth wave peaked in January 2022. [6] The present study focuses on the fourth wave, which occurred in mid-January 2022 and reached a maximum mortality rate of 282 deaths per day on 28 January. The Colombian National Institute of Health (INS) reported that 80% of these deaths were attributed to the Omicron variant [7,8], which was designated as a VOC by the WHO on 26 November 2021 [9]. This variant has a large number of mutations, with 32 affecting the Spike protein. According to the Centers for Disease Control and Prevention (CDC) in the United States, despite its increased transmissibility, the Omicron variant is associated with less severe clinical presentations [10].

In Colombia, the national vaccination plan against COVID-19 began on 17 February 2021. One year later, on 17 February 2022, 33.06 million individuals were fully vaccinated (58.1%), and 7,784,212 had a booster dose [11,12,13]. Considering the development of the national vaccination process in Colombia, one of the strategies used to address vaccine shortage and production delays was combining vaccines from different pharmaceutical companies. This heterologous vaccination approach, supported by increasing scientific evidence and international organizations, involves administering a vaccine platform different to the one used for the primary vaccine series, for the completion of the series [14]. Heterologous induction and booster strategies can offer immunological advantages in terms of the breadth and longevity of protection provided by currently available vaccines, generating robust levels of antibodies against the COVID-19 virus and a greater T-cell response and memory B-cell response, compared to a homologous viral vector vaccine regimen [2,3]. Additionally, this approach simplifies the vaccine administration logistics because it allows for more flexible vaccine administration in environments with limited or unpredictable supply [15]

The booster strategy began in October 2021 for adults older than 70 years old and patients with a history of organ transplant, but it was rapidly expanded to include adults with other risk factors and healthcare personnel. By February 2022, the booster was accessible to the entire adult population [16].

As of 2 June 2023, Colombia had administered 90,506,612 vaccine doses, with 37.019 million people completing the primary vaccine series yet, despite these efforts, only 14.792 million individuals had received a booster dose.

The future impact of the SARS-CoV-2 virus will be determined by a complex interaction of factors, including population protection mediated by natural immunity (recovery rate) and acquired immunity (vaccination rate), the availability of new medications and interventions, the biological behavior of new variants, and population behavior. Therefore, it is essential to study each of these factors and their impact on the course of the pandemic. Several studies have evaluated and confirmed the effectiveness of vaccination in various populations, particularly in terms of reducing serious outcomes such as hospitalization and death [17,18,19,20]. Additionally, some studies have investigated the effectiveness of vaccination against infection, identifying the need for booster doses. [21]

This study aimed to explore the effectiveness of vaccination schedules on the probability of infection, adjusted to other variables such as age, gender, nationality, occupation, socioeconomic status, health regimen, symptoms, medical history, and vaccination information.

## 2. Materials and Methods

### 2.1. Study Design and Participants

We conducted a cross-sectional study that included individuals who underwent a RT-PCR test for COVID-19 detection at a specific laboratory in Bogotá, Colombia, between 30 December 2021 and 7 February 2022. A total of 1468 subjects was included in the study.

### 2.2. Data Collection

Data were collected retrospectively using the mandatory national notification form, which was filled out by every individual who underwent a RT-PCR test for the diagnosis of SARS-CoV-2 infection at the time of sampling. However, these forms did not include the reasons for the PCR testing.

Only cases that provided information about their vaccination status were included in the study. Demographics, clinical, and vaccination variables were included in the final database. The result of the RT-PCR for SARS-CoV-2 infection was considered the dependent variable and categorized as positive or negative based on the test result.

Independent variables included socio-demographic factors, clinical variables and vaccination status. Vaccination status was categorized as complete (two doses of BNT162b2, CoronaVac, mRNA-1273 or ChAdOx1 vaccines, or one dose of Ad26.COV2.S vaccine) or incomplete, according to Colombian regulations. Booster vaccination was defined as the receipt of an additional booster dose after completing the recommended vaccination regimen [22].

We also considered health insurance affiliation. The Colombian health system is universal, and all citizens must join one of three insurance schedules: a contributory plan for formal employees and self-employed workers with the capacity to pay; a subsidized plan for informal workers and low-income individuals who are unable to pay; and a special plan for teachers and military personnel. Only 5% of the population is uninsured.

### 2.3. Data Analysis

Both the dependent and independent variables obtained from the national notification forms were analyzed using SPSS version 21 software [23]. Descriptive statistics were calculated for the variables, and simple and multiple logistic regressions were performed to correlate the COVID-19 test results with vaccination status, taking into account sociodemographic and clinical variables.

### 2.4. Ethical Aspects

This study was approved by an ethics committee with the code DVO005 1973-CV1560. The research was conducted without any risk (Resolution 8430 of 1993), as it employed a retrospective observational methodology without any intervention on the research subjects. Prior to sample collection, participants provided a signed informed consent, granting permission for the use of their information and test results for research purposes. The data were protected according to law 1581 of 2012.

## 3. Results

This study included 1468 participants, of whom 36.6% (n = 538) were confirmed to have COVID-19 via PCR. The socioeconomic characteristics of the participants are described in Table 1. The sample included individuals ranging from 1 to 93 years old, with a mean of 39.19 years (SD 16.05). Among the participants, 57.9% (n = 850) were women, and 92.4% (n = 1356) were from Colombia, predominantly from Bogotá (67.7%, n = 994). In terms of occupation, 26.4% (n = 387) identified themselves as having a high level of education, and 33.70% (n = 495) reported to be employees. Additionally, 29.8% (n = 438) of the participants belonged to the middle economic level. Furthermore, 23.6% (n = 347) of the participants reported to have traveled 14 days prior to the onset of symptoms. This time frame was established for epidemiologic purposes at the time of the pandemic.

Table 2 presents the clinical characteristics of the participants. Only 1.1% of the included records pertained to hospitalized individuals. Among the reported symptoms, the most common were cough 17.7% (n = 550), odynophagia 15.5% (n = 437), and headache 11.9% (n = 336). Regarding the comorbidities, hypertension was the most common, affecting 5.3% of the participants (n = 82).

Of the individuals, 97.7% were vaccinated (n = 1434), and 90.3% (n = 1326) had completed the full vaccination schedule. However, only 20.8% (n = 306) of vaccinated individuals had completed the vaccination schedule with a booster dose. The most frequently administered vaccine among the population was BNT162b2, accounting for 30.9% (n = 467) of the cases (Table 3).

In the multivariate analysis, lower infection rates were associated with the booster dose compared to the complete vaccination schedule (OR = 0.75 CI 0.58–0.99) (Table 4), while higher infection rates were associated with age and lower socio-economic status (Table 5).

## 4. Discussion

This study provides evidence that the booster dose is the most effective vaccination schedule for reducing the COVID-19 infection probability. According to our results, people with a booster dose had 25% reduced odds of developing a COVID-19 infection.

Our findings support the need to implement the booster vaccination as a measure to reduce COVID-19 infection rates (OR = 0.75, CI 0.58–0.99). The data collection for this study was conducted during the midst of the fourth wave of COVID-19 in Colombia, with the predominant circulating variant being Omicron. A systematic review and meta-analysis conducted by Chenchula et al. revealed that the risk of reinfection by Omicron was significantly higher than that of previous variants. In addition, they demonstrated a decrease in overall protection against COVID-19 infection among individuals with two-dose vaccinations. The booster dose showed a significant increase in neutralizing immunity and higher levels of antibody titers against Omicron, resulting in lower rates of symptomatic infection [24].

It is worth noting that the circulating lineages at the time of the study are an important factor that could impact our results. As previously mentioned, during the time frame when the sample was collected, the Omicron variant was predominant in Colombia, and by the same period [25], the administered vaccines were monovalent, mainly targeting the original strain of the virus with proven coverage against the Alpha and Delta variants [3].

According to Lauring et al., in the United States winter of 2021–2022, three doses of an mRNA monovalent vaccine were necessary to achieve effectiveness against the Omicron variant, while only two doses achieved the same results for Alpha and Delta variants earlier in the same year. This evidence overlaps with the time period evaluated by the authors, leading them to consider that these results could also be explained by the type of vaccines used at the time the sample was collected [3].

Additionally, there are different studies showing that the Omicron variant is related to a decrease in vaccine effectiveness against COVID-19 infection. Andrews et al. conducted a study which showed that the effects of the vaccine 20 weeks after the second dose waned rapidly; therefore, the booster shot substantially increased its effectiveness. The study concludes that a complete schedule of two doses was not enough to protect against the Omicron variant’s mild infection, and even with the booster shot, the effectiveness diminished over time [26].

The WHO‘s Strategic Advisory Group of Experts on immunization (SAGE) currently explicitly recommends a booster dose for medium- and high-risk priority-use groups. According to their most recent statement, there are three priority-use groups for COVID-19 vaccination: high, medium, and low, based on the severe disease and death risks. The high priority group includes older adults, younger adults with significant comorbidities, people with immunocompromising conditions, pregnant women, and frontline health workers. The medium-priority group comprises healthy adults under 50–60 years without comorbidities and children and adolescents with comorbidities. The low-priority group includes healthy children and adolescents between 6 months and 17 years [27]. For people outside these categories, countries have established different recommendations based on factors such as the national disease burden, cost effectiveness, and epidemiological situation, among others.

Expanding booster recommendations to other socio-demographic groups, in addition to those already recommended by SAGE, could be considered in the Colombian population, since only 20.8% of our individuals received the booster dose, and considering that the mean age in our population study was 39 years old and 55.7% were healthy participants.

At the time patients were recruited for this study, the Colombian National Vaccination Plan had been in place for 10 months, showing a significant reduction in the number of severe and fatal cases of SARS-CoV-2 infection. By February 2022, it was estimated that 7,784,212 people had received a booster dose, 27,057,045 had completed the vaccination schedule, and 35,047,468 had received only the first or the only dose vaccination. This may explain the decrease in mortality rates despite the high number of cases observed during the fourth peak when this study took place [13]. These data, of course, are not limited to Colombia, as it has been widely reported that the COVID-19 vaccination played a major role in the final and successful control of the pandemic worldwide [28].

From a biological point of view, the effectiveness of the booster dose could rely on the previously reported seroconversion rates of antibodies against the spike protein and the receptor binding-domain, as well as the neutralization antibody by plaque reduction neutralization tests, achieved 1 and 6 months after administration of the booster shot [29]. Furthermore, it has been also been reported that this seroconversion includes the increase in antibodies against specific SARS-CoV-2 lineages like Delta and Omicron variants, specially 6 months after the booster [30,31].

Additionally, Vadrevu et al. reported an increase in the Th1:Th2 and the media IgG-secreting memory B cells when comparing populations with and without the booster, while Chiu et al. reported an enhancement of the cellular immune response for the Delta variant and the wildtype when receiving a booster dose [30,32].

Now that the bivalent COVID-19 vaccines are available in some countries but not currently in Colombia, additional studies following similar methodologies to the one performed in this study should be conducted to confirm the necessity of a third dose with the use of COVID-19 ‘second generation’ vaccines. In addition, the Omicron variant has been described as a milder disease-causing variant of the SARS-CoV-2 infection, when compared to Alpha and Delta lineages. These differences should be taken into account as confounding factors when conducting further studies, highlighting the fact that controlling the Omicron or other emerging variants goes beyond reducing clinically severe COVID-19 infection, as it also contributes to preventing the emergence of other VOCs.

Despite this study not intending to compare the effectiveness of vaccinations among different vaccine types, we found a higher infection rate in participants who received CoronaVac (OR = 1.518, CI 1.079–2.135) and Ad26.COV2.S (OR = 1.691, CI 1.137–2.514). Similarly, Chi et al. previously reported that the efficacy against COVID-19 infection was 91.3% for BNT162b2, 93.2% for mRNA-1273, 74% for ChAdOx1, 52.4% for Ad26.COV2.S, and 50.7% for CoronaVac [33]. Additionally, a comparison of the probability of each intervention being better than all competing interventions (P-score) made among different COVID-19 vaccines revealed that BNT162b2 (P-score: 0.952) and mRNA-1273 (P-score: 0.843) were classified with the highest probability of efficacy against symptomatic COVID-19. On the other hand, CoronaVac (P-score: 0.570) and Ad26.COV2.S (P-score: 0.198) showed lower probabilities of efficacy against the infection. [34].

It is known that countries with lower socioeconomic levels have higher transmission and mortality rates. Our study identified a significant association between lower socioeconomic status and a higher risk of infection. This could be related to different health determinants such as living situations, education, and health practices, among others. Although the functionality of a society and its socioeconomic determinants in health may differ between nationalities, a study by Hawinks et al. in the United States analyzed the number of cases and fatalities, pairing them with the county level of the distressed communities index to measure socioeconomic status and determinant factors. The results showed that despite a higher rate of infection among higher-income people, other factors such as a lack of high school education and a higher proportion of black Americans were the strongest associations with transmission [35].

Despite the high percentage of complete vaccinations in the studied sample, we also found an association between age and infection, identifying higher rates of COVID-19 infection in older people. It is well known that older people have a lower rate of vaccine response [36]. A previous study in Colombia found that COVID-19 vaccine effectiveness decreased with age, regardless of the type of vaccine used, being lower in traditional vaccine platforms [37]. Older people have also shown a greater and faster decline in neutralizing antibodies, making the vaccination booster more advisable [38].

In 2021, the Ministry of Health and Social Protection in Colombia conducted a population-based cohort study in adults aged 60 and older. The study aimed to estimate the effectiveness of COVID-19 vaccines in preventing hospitalization and death among older adults with a complete vaccination schedule during the first 5 months of the National COVID-19 Vaccination Plan in Colombia. The study found that the effectiveness of vaccines in this population group was 69.9% in preventing non-fatal hospitalization, 79.4% in preventing death from COVID-19 after hospitalization, and 74.5% in preventing death from COVID-19 among those who were not hospitalized [39].

On the other hand, elderly vaccination has proven to be safe and effective, especially after multiple doses. Adverse events are lower than in young adults. Despite immunosenescence, vaccine adjuvants enhance humoral response and increase serum conversion [40]. A higher incidence of infection, mortality, and serious outcomes has been reported throughout the pandemic in the elderly. Multiple studies have found that people over 60 years of age were the most affected [25,37,40]. Protection of the older population has been demonstrated to be critical in the prevention and control of the pandemic, and the WHO has strongly prioritized vaccination efforts for this group [41].

Some limitations of this study were the population size of a convenience sample and the lack of epidemiological data to further identify the different homologous and heterologous vaccination schedules in order to analyze their different effectiveness. Therefore, the study focuses on those factors that could show a statistical significance. Additionally, some of the socioeconomic and demographic data could be expanded to better understand the health determinants of the population, so that they could be addressed in the future. Lastly, this was a cross-sectional study that did not follow up with the studied population, so it is not certain how the effectiveness of the booster dose schedule may change over time.

## 5. Conclusions

The booster vaccination schedule has demonstrated effectiveness as a strategic measure against COVID-19 infection, surpassing the results achieved through the administration of a complete initial vaccination schedule. Consequently, it is imperative to enhance efforts in order to administer booster vaccines to the general population and raise awareness about their critical significance. Furthermore, it is crucial to intensify research endeavors to determine the duration of effectiveness and the necessity for additional booster doses.

## Figures and Tables

**Table 1 vaccines-11-01461-t001:** Socioeconomic characteristics.

Variable		Total n = 1468		Positive COVID-19 Cases n = 538		Negative COVID-19 Cases n = 930	
		Frequency	Percentage	Frequency	Percentage	Frequency	Percentage
**Age**		Min 1–Max 93		Min 2–Max 92		Min 1–Max 93	
		Mean 39.19 (SD 16.05)					
**Nationality**	**Other**	112	7.60%	22	1.50%	90	6.10%
	**Colombian**	1356	92.40%	516	35.10%	840	57.20%
**Origin**	**Other**	474	32.30%	230	15.70%	244	16.60%
	**Bogotá**	994	67.70%	308	21.00%	686	46.70%
**Gender**	**Male**	618	42.10%	213	14.50%	405	27.60%
	**Female**	850	57.90%	325	22.10%	525	35.80%
**Occupation**	**Housewife**	123	8.40%	46	3.10%	77	5.20%
	**Salesman**	40	2.70%	9	0.60%	31	2.10%
	**Freelance**	72	4.90%	25	1.70%	47	3.20%
	**Health professional**	85	5.80%	37	2.50%	48	3.30%
	**High level education**	387	26.40%	120	8.20%	267	18.20%
	**Student**	141	9.60%	44	3.00%	97	6.60%
	**Employee**	495	33.70%	214	14.60%	281	19.10%
	**Does not apply**	125	8.50%	43	2.90%	82	5.60%
**Economic status**	**Low**	383	26.10%	183	12.50%	200	13.60%
	**Medium**	438	29.80%	160	10.90%	278	18.90%
	**High**	144	9.80%	46	3.10%	98	6.70%
	**Unknown**	503	34.30%	149	10.10%	354	24.10%
**Health regimen**	**Contributory**	965	65.70%	378	25.70%	587	40.00%
	**Uninsured**	66	4.50%	9	0.60%	57	3.90%
	**Special**	11	0.70%	5	0.30%	6	0.40%
	**Subsidized**	54	3.70%	25	1.70%	29	2.00%
	**Undetermined**	313	21.30%	102	6.90%	211	14.40%
	**Foreign**	59	4.00%	19	1.30%	40	27.00%

**Table 2 vaccines-11-01461-t002:** Clinical Characteristics.

Variable			Total n = 1468		Positive COVID-19 Cases n = 538		Negative COVID-19 Cases n = 930	
			Frequency	Percentage	Frequency	Percentage	Frequency	Percentage
**Hospitalized**			16	1.10%	9	0.60%	7	0.50%
**Symptoms**	**Some symptoms**	**Cough**	500	12.63%	274	6.92%	226	5.70%
		**Fever**	212	5.37%	133	3.36%	79	2.00%
		**Odynophagia**	437	11.00%	245	6.17%	192	4.82%
		**Dyspnea**	76	1.90%	44	1.10%	32	0.79%
		**Fatigue/adynamia**	156	3.96%	89	2.26%	67	1.69%
		**Rhinorrhea**	240	6.07%	107	2.70%	133	3.36%
		**Conjunctivitis**	37	0.92%	22	0.54%	15	0.37%
		**Headache**	336	8.46%	189	4.76%	147	3.70%
		**Diarrhea**	78	1.95%	38	0.94%	40	1.00%
		**Anosmia**	25	0.65%	15	0.39%	10	0.26%
		**Others**	52	1.30%	29	0.72%	23	0.57%
		**Total**	796	54.20%	384	26.20%	412	28.10%
	**Asymptomatic**		512	34.90%	127	8.70%	385	26.20%
	**Unknown**		160	10.90%	27	1.80%	133	9.10%
**Medical History**	**Any**	**Asthma**	39	2.12%	16	0.86%	23	1.25%
		**COPD**	6	0.32%	2	0.10%	4	0.21%
		**Diabetes**	35	1.91%	16	0.87%	19	1.03%
		**HIV**	3	0.17%	3	0.17%	0	0,0%
		**Heart disease**	13	0.71%	7	0.38%	6	0.32%
		**Cancer**	21	1.15%	6	0.32%	15	0.82%
		**Obesity**	24	1.32%	8	0.43%	16	0.87%
		**Renal insufficiency**	8	0.44%	2	0.33%	6	0.11%
		**Immunosuppression**	4	0.21%	1	0.05%	3	0.15%
		**Smoker**	32	1.74%	10	0.54%	22	1.19%
		**Hypertension**	82	4.47%	34	1.85%	48	2.61%
		**Tuberculosis**	1	0.06%	0	0.00%	1	0.06%
		**Others**	82	4.47%	36	1.96%	46	2.50%
		**Total**	280	19.10%	112	7.60%	168	11.40%
	**No Medical History**		818	55.70%	324	22.10%	494	33.70%
	**Unknown**		370	25.20%	102	6.90%	268	18.30%
**Recent travel**			347	23.60%	139	9.50%	208	14.20%

COPD Chronic Obstructive Pulmonary Disease; HIV Human Immunodeficiency Virus.

**Table 3 vaccines-11-01461-t003:** Vaccination Characteristics.

Variable		Total Population n = 1468		Positive COVID-19 Cases n = 538		Negative COVID-19 Cases n = 930	
		Frequency	Percentage	Frequency	Percentage	Frequency	Percentage
**Vaccinate**		1434	97.70%	524	35.70%	910	62.00%
**Booster vaccination schedule**		306	20.80%	97	6.60%	209	14.20%
**Complete vaccination schedule**		1326	90.30%	492	33.50%	834	56.80%
**Vaccine dose**	**1**	231	15.70%	91	6.20%	140	9.50%
	**2**	913	62.20%	342	23.30%	571	38.90%
	**3**	278	18.90%	85	5.80%	193	13.10%
	**4 or more**	3	0.20%	2	0.10%	1	0.10%
**Vaccine type**	**BNT162b2**	467	30.90%	160	10.59%	307	20.30%
	**CoronaVac**	325	21.50%	135	8.92%	190	12.57%
	**mRNA-1273**	281	18.60%	86	5.69%	195	12.90%
	**ChAdOx1**	203	13.40%	74	4.89%	129	8.50%
	**Ad26.COV2.S**	182	12.00%	79	5.20%	103	6.79%
	**Other**	10	0.70%	1	0.07%	9	0.63%
	**Unknown**	43	2.80%	17	1.10%	26	1.69%

**Table 4 vaccines-11-01461-t004:** Binary logistic regressions of COVID-19 infection on vaccination status.

Variable	Category	OR	95% C.I.		*p* Value
			Lower	Upper	
**Vaccinate**	**Yes**	0.82	0.41	1.64	0.580
	**No**	Comparison category			
**Complete vaccination schedule**	**Yes**	1.23	0.85	1.78	0.269
	**No**	Comparison category			
**Booster vaccination schedule**	**Yes**	0.75	0.58	0.99	0.044
	**No**	Comparison category			

**Table 5 vaccines-11-01461-t005:** Binary logistic regression of COVID-19 infection on booster vaccination schedule.

Variable	Category	OR	95% C.I.		*p* Value
			Lower	Upper	
**Booster vaccination schedule**	**Yes**	0.73	0.54	0.99	0.045
	**No**	Comparison category			
**Age**		1.00	1.00	1.01	0.029
**Economic status**	**Low**	2.02	1.54	2.65	0
	**Medium**	1.28	0.99	1.67	0.057
	**High**	Comparison category			
**Health Professional**	**Yes**	1.50	0.95	2,3	0.08
	**No**	Comparison category			
**Student**	**Yes**	1.02	0.66	1.58	0.909
	**No**	Comparison category			
**Vaccine type**	**BNT162b2**	1.15	0.83	1.59	0.376
	**CoronaVac**	1.49	1.06	2.10	0.019
	**mRNA-1273**	Comparison category			
	**ChAdOx1**	1.17	0.79	1.71	0.418
	**Ad26.COV2.S**	1.63	1.10	2.43	0.014
	**Other**	0.25	0.03	2.06	0.200
	**Unknown**	1.30	0.66	2.57	0.439

## Data Availability

Additional data related to this paper may be requested from the authors.

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
