# Peer review of "Effectiveness of the Booster Dose in Protecting against COVID-19, Colombia 2022"

_vaccines, 2023, doi:10.3390/vaccines11091461_

Round 1
Reviewer 1 Report
The manuscript (ID: vaccines-2510487) aimed to explore the effectiveness of a booster dose regimen compared to other vaccination statuses (incomplete or complete ) in terms of the probability of infection in Colombia.
Comments:
- In the Abstract, state what was the indication for conducting the PCR test in this study.
- In the Abstract, indicate the number (%) of participants who were PCR positive in this study.
- In the Introduction section, the epidemiological situation related to the COVID-19 pandemic and vaccination against COVID-19 is correctly described, with a detailed description of the circumstances and course of the pandemic in Colombia, with special reference to the national vaccination process in Colombia.
- Also, the objectives of the study are defined at the end of the Introduction section. The authors should modify the last sentence in the Introduction section, because the paper does not show only the variables age and gender.
- Does `WHO-designated VOC` represent `WHO-designated variants of concern (VOC)`? Define abbreviation.
- In the Materials and Methods chapter, Study design and Participants, Data collection (dependent and independent variables are defined), Data analysis and Ethical aspects are in detail stated.
- Note: In the subsection Data collection, state what was the indication for conducting the PCR test in this study.
- Comments for the Results section:
- To specify which Table shows the results listed in the first sentence of this chapter, I quote `This study included 1468 participants, of whom 36.6% (n=538) were confirmed to have COVID-19 infection via PCR.`.
- Where are the data for `participants reported having traveled prior to taking the COVID 19 test` shown on Table 1 (or in general in the entire paper)? Define `travel`.
- In the titles of all Tables in the work, the number of participants to which the results refer should be indicated.
- Reconstruct the results so that Tables 1 and 2 show separately the results for PCR positive and PCR negative participants. Determine the statistical significance between PCR positive and PCR negative participants according to all described characteristics/variables.
- Table 4 is not clear enough. Correct this.
- The results did not correspond with the set goals.
- Correct the Results chapter as a whole, in such a way as to take into account the goals set in this work.
- The Discussion section does not correspond well with the data presented in the Results section.
The quality of English language is appropriate.
Author Response
Bogotá, July 25th 2023
Editor-in-Chief
Vaccines Journal
Dear Editorial Staff,
On behalf of myself and the co-authors, I would like to express sincere appreciation for the opportunity to submit our article entitled "Effectiveness of the booster dose in protecting against COVID-19, Colombia 2022" for review in your journal. Additionally, we are deeply grateful for the invaluable comments and suggestions provided by the reviewers, as they have significantly enhanced the quality of our work.
Below, we present the changes made in response to the reviews:
Reviewer 1:
- Comment: The word count (main text) of your manuscript is 3449, and we would appreciate it if you could increase to over 4000 words.
Answer: New paragraphs have been included, allowing the total word count of the manuscript to exceed 4000
- Comment: Please sign the enclosed the Disclosure of Potential Conflicts of Interest form.
Answer: The Disclosure of Potential Conflicts of Interest form was completed by all the authors and is attached as an annex to this letter.
- Comment: Please provide us with the name of ethic committee.
Answer: The project was reviewed and approved by Research Ethics Committee of Universidad del Rosario in Colombia
- Comment: Please check that all references are relevant to the contents of the manuscript.
Answer: It has been confirmed that all references included in the manuscript are pertinent and relevant to its content.
- Comment: In the Abstract, state what was the indication for conducting the PCR test in this study.
Comment: In the subsection Data collection, state what was the indication for conducting the PCR test in this study.
Answer: The data used for this study were sourced from mandatory notification records established by the Colombian national government, which, regrettably, did not include the indications for the PCR testing. As a result, this information was not incorporated into the paper. Moreover, we firmly believe that such details are not essential for achieving the objectives of the manuscript. To address this issue, we have provided a clarification in the data collection section.
- Comment: In the Abstract, indicate the number (%) of participants who were PCR positive in this study.
Answer: This information was included in the abstract “of whom 36.6% (n=538) had a positive PCR test for COVID-19”
- Comment: Also, the objectives of the study are defined at the end of the Introduction section. The authors should modify the last sentence in the Introduction section, because the paper does not show only the variables age and gender.
Answer: All the variables used in the development of the article have been included in the indicated sentence “Additionally, the study seeks to describe the behavior of cases according to other variables such as age, gender, nationality, occupation, socioeconomic status, health regimen, symptoms, medical history, and vaccination information.”
- Comment: Does `WHO-designated VOC` represent `WHO-designated variants of concern (VOC)`? Define abbreviation.
Answer: The abbreviation was defined as Variants of Concern (VOCs)
- Comment: To specify which Table shows the results listed in the first sentence of this chapter, I quote `This study included 1468 participants, of whom 36.6% (n=538) were confirmed to have COVID-19 infection via PCR.
Comment: Reconstruct the results so that Tables 1 and 2 show separately the results for PCR positive and PCR negative participants. Determine the statistical significance between PCR positive and PCR negative participants according to all described characteristics/variables.
Answer: Tables 1, 2, and 3 were disaggregated into results for PCR positive and negative cases to facilitate results comprehension and provide information in the most suitable manner
- Comment: Where are the data for `participants reported having traveled prior to taking the COVID 19 test` shown on Table 1 (or in general in the entire paper)? Define `travel`
Answer: The data included in the article originates from mandatory notification records for COVID-19, which incorporate the variable 'recent travel' since it was of particular interest during the early stages of the pandemic. This variable refers to any travel within the 14 days prior to the onset of symptoms. The definition of this variable was included in the Results section.
- Comment: In the titles of all Tables in the work, the number of participants to which the results refer should be indicated.
Answer: In the titles of Tables 1, 2, and 3, the number of participants to which the results refer has been included. The total values refer to the entirety of the study participants, n=1468, with the values disaggregated for PCR positive representing 538 individuals and PCR negative representing 930 individuals.
- Comment: Table 4 is not clear enough. Correct this.
Answer: Table 4 presents the results of a binary logistic regression conducted to assess the association of COVID-19 infection with variables such as booster vaccination status and other variables of interest. The table has been revised to enhance its comprehensibility.
- Comment: The results did not correspond with the set goals.
Comment: Correct the Results chapter as a whole, in such a way as to take into account the goals set in this work.
Comment: The Discussion section does not correspond well with the data presented in the Results section.
Answer: The entire article was thoroughly reviewed by all the authors to make the necessary changes and ensure that the objective, results and discussion were in good concordance. The order of the discussion was restructured to highlight the importance of our main result and mantain consistency throughout the article.
We hope that these revisions meet your expectations and align with the quality standards of your journal. We are confident that the results of our study will be of great interest to the scientific community and will contribute significantly to the global understanding of the effectiveness of booster doses against COVID-19.
Once again, we sincerely appreciate the opportunity to consider the publication of our work in Vaccines and eagerly look forward to receiving positive news regarding our manuscript.
Sincerely,
Ángela María Ruiz Sternberg
Corresponding author

Reviewer 2 Report
In this manuscript, the authors performed a review and analysis of the COVID19 cases in Columbia in early 2022 and the effect of a booster COVID19 vaccination of the infections rate among tested individuals.
Some of the main comments I would like the authors to address are:
1- Could there be an association between the person getting the booster shot and seeking testing more frequently that non-boosted subjects? In other words, those who get boosted are obviously more conscious of there health, and probably would seek testing more often than non-boosted. Also, given the mild nature of the omicron variant, many infected people may not seek testing, even if they’ve shown symptoms. The reason I wanted to address this point is to ensure there was no bias in the data collected and analyzed.
2- In table 3, it seems that having any of the vaccines was associated with much higher rates of infection. Could you comment on that?
3- From the analysis you performed, and when comparing the different vaccines, did any of the vaccines performed better than the others? This might be important as another factor in preventing infection, even in non-boosted subjects.
Minor comments:
1- For table 1, please use lines to separate the different categories, or use a different format, as the current format is confusing (difficult to separate the variables form each other). The same applies to table 3. Table 2 is ok.
2- On page 2, you stated regarding the heterologous booster approach that “Additionally, this approach simplifies the vaccine administration logistics”. Would you please explain why the heterologous vaccine will simplify the vaccine administration, and how that would be justified if there is no relevant clinical data to support the combination?
3- There have been multiple papers that study the effect of boosters on COVID19 infection. Although the results differ a bit from what you’ve shown, but I think the data supports your conclusions that a booster may benefit recipients’ in reducing COVID19 infection cases.
I suggest the manuscript go through one round of English editing to imrove readability..
Author Response
Bogotá, July 25th 2023
Editor-in-Chief
Vaccines Journal
Dear Editorial Staff,
On behalf of myself and the co-authors, I would like to express sincere appreciation for the opportunity to submit our article entitled "Effectiveness of the booster dose in protecting against COVID-19, Colombia 2022" for review in your journal. Additionally, we are deeply grateful for the invaluable comments and suggestions provided by the reviewers, as they have significantly enhanced the quality of our work.
Below, we present the changes made in response to the reviews:
Reviewer 2:
- Comment: Could there be an association between the person getting the booster shot and seeking testing more frequently that non-boosted subjects? In other words, those who get boosted are obviously more conscious of there health, and probably would seek testing more often than non-boosted. Also, given the mild nature of the omicron variant, many infected people may not seek testing, even if they’ve shown symptoms. The reason I wanted to address this point is to ensure there was no bias in the data collected and analyzed.
Answer: Within the data collected in this project, the reasons and indications for taking the PCR test were not considered. Consequently, we cannot ascertain whether individuals took the test for self-care reasons or whether this could be associated with the administration of the booster dose. We believe that this observation goes beyond the objectives of the study, and since we lack sufficient evidence to support it, we have not included it in the manuscript.
- Comment: In table 3, it seems that having any of the vaccines was associated with much higher rates of infection. Could you comment on that?
Answer: Table 3 presents the vaccination characteristics of the participants, including the total number of individuals vaccinated with each vaccine brand, the vaccination schedule, and the number of doses administered. The rate of infection was not included in this table. To make it clearer, the table has been disaggregated to present the total population against the positive and negative cases.
- Comment: From the analysis you performed, and when comparing the different vaccines, did any of the vaccines performed better than the others? This might be important as another factor in preventing infection, even in non-boosted subjects.
Answer: We observed a higher infection rate in participants who received CoronaVac (OR=1.518, CI 1.079 - 2.135) and Ad26.COV2.S (OR=1.691, CI 1.137 - 2.514). These findings are consistent with those of other authors and have also been included in the discussion section of the manuscript. However, it is important to note that the main objective of this study did not include analyzing the comparison of effectiveness between vaccine brands. Therefore, this result should be considered carefully and interpreted in light of the study’s primary focus.
- Comment: For table 1, please use lines to separate the different categories, or use a different format, as the current format is confusing (difficult to separate the variables form each other). The same applies to table 3. Table 2 is ok.
Answer: The separation format of variables in Tables 1, 2, and 3 was modified to improve its comprehension.
- Comment: On page 2, you stated regarding the heterologous booster approach that “Additionally, this approach simplifies the vaccine administration logistics”. Would you please explain why the heterologous vaccine will simplify the vaccine administration, and how that would be justified if there is no relevant clinical data to support the combination?
Answer: The application of heterologous schemes allows for a flexible approach to vaccination dosing in places where there might be issues with vaccine supply, particularly in developing countries like Colombia. This clarification has been added to the text and referenced with the guidelines provided by the World Health Organization (WHO) for the application of heterologous vaccination schemes.
- Comment: There have been multiple papers that study the effect of boosters on COVID19 infection. Although the results differ a bit from what you’ve shown, but I think the data supports your conclusions that a booster may benefit recipients’ in reducing COVID19 infection cases.
Answer: We identified the idea referring to the lack of studies on the effectiveness of vaccination against infection, and we clarified that such studies demonstrate the need for booster doses to maintain effectiveness over time.
- Comment: I suggest the manuscript go through one round of English editing to improve readability.
Answer: The article has undergone peer review and editing by academics with a strong command of the English language.
We hope that these revisions meet your expectations and align with the quality standards of your journal. We are confident that the results of our study will be of great interest to the scientific community and will contribute significantly to the global understanding of the effectiveness of booster doses against COVID-19.
Once again, we sincerely appreciate the opportunity to consider the publication of our work in Vaccines and eagerly look forward to receiving positive news regarding our manuscript.
Sincerely,
Ángela María Ruiz Sternberg
Corresponding author

Round 2
Reviewer 1 Report
The authors have satisfactorily responded to all my comments and made appropriate changes in the revised version of this paper. Overall, the work is clearer and more informative in the context of the important topic it deals with. I thank the authors.
The quality of English language is appropriate.